# Black to the Future: Making the Case for Indigenist Health Humanities

**DOI:** 10.3390/ijerph18168704

**Published:** 2021-08-18

**Authors:** Chelsea Watego, Lisa J. Whop, David Singh, Bryan Mukandi, Alissa Macoun, George Newhouse, Ali Drummond, Amy McQuire, Janet Stajic, Helena Kajlich, Mark Brough

**Affiliations:** 1School of Public Health and Social Work, Queensland University of Technology, Brisbane 4059, Australia; m.brough@qut.edu.au; 2National Centre for Epidemiology and Population Health, Australian National University, Canberra 2601, Australia; lisa.whop@anu.edu.au; 3Aboriginal and Torres Strait Islander Studies Unit, The University of Queensland, Brisbane 4067, Australia; david.singh@uq.edu.au; 4School of Languages and Cultures, The University of Queensland, Brisbane 4067, Australia; b.mukandi@uq.edu.au; 5School of Justice, Queensland University of Technology, Brisbane 4000, Australia; alissa.macoun@qut.edu.au; 6Macquarie Law School, Macquarie University, Sydney 2109, Australia; georgen@justice.org.au; 7School of Nursing, Queensland University of Technology, Brisbane 4059, Australia; ali.drummond@qut.edu.au; 8School of Political Science and International Studies, The University of Queensland, Brisbane 4067, Australia; a.mcquire@uq.edu.au (A.M.); h.kajlich@uq.edu.au (H.K.); 9Faculty of Medicine, The University of Queensland, Brisbane 4067, Australia; j.stajic@uq.edu.au

**Keywords:** Indigenous, humanities, race, sovereignty, health, justice, Indigenist, transdisciplinarity

## Abstract

This paper outlines the development of Indigenist Health Humanities as a new and innovative field of research building an intellectual collective capable of bridging the knowledge gap that hinders current efforts to close the gap in Indigenous health inequality. Bringing together health and the humanities through the particularity of Indigenous scholarship, a deeper understanding of the human experience of health will be developed alongside a greater understanding of the enablers to building a transdisciplinary collective of Indigenist researchers. The potential benefits include a more sustainable, relational, and ethical approach to advancing new knowledge, and health outcomes, for Indigenous people in its fullest sense.

## 1. Introduction

*Rather than going beyond conflict towards a more humane world, we seem to be going in the opposite direction. People seem more than ever to be polarised along lines of difference, more seeking to exploit difference to divide rather than cooperate...I’m not suggesting that we go back to the past: but that we might all draw hope from the Murri experience, and learn from it, about what it might be possible to achieve in the future. Seeking co-operation out of conflict can be a first step along the road. It will take time. But drawing on our experience as Murris, while we don’t expect to see dramatic change in a life-time, we know change is possible. We see our future stretching out as far in front of us as it does behind us. And we hope that our contribution to the process will be recognised and valued. (Dr Lilla Watson, Gungalu and Birra Gubba)* [1] (Murri is a term used by Aboriginal people to describe themselves and is used most typically in Queensland and north-west New South Wales regions.)

This is not a traditional Indigenous health research paper. There is no specific health problem that we propose to solve via our knowing, whether through canvassing the available literature or via a discrete study involving Indigenous subjects. We present no alarming statistics, or clinical interventions. This paper is instead a story, a story of the emergence of a new field of research from so-called Australia which redefines the parameters by which we understand health and humanity via a foregrounding of Indigenous sovereignty, both locally and globally. Rather than be positioned as peoples destined to die, or researched to death, Indigenist Health Humanities as a field of research is based on an Indigenous imagining of ‘a future stretching out as far in front of us as it does behind us’ [1].

We tell this story as a multidisciplinary team of Indigenous and non-Indigenous academics which comprises Aboriginal health workers as well as a nurse, epidemiologist, critical social scientist, journalist, philosopher, political scientist, lawyer, and a critical race scholar. Each of us is concerned with Indigenous health and wellbeing in its broadest sense, encompassing the historical, political, cultural, social, and economic conditions of Indigenous life worlds. As such, we tell this story in a different kind of way than the traditional health research team would, and, while we represent a research team that was recently awarded an Australian Research Council grant to establish a new field of research, we too represent a community brought together through a shared values framework, namely the Inala Manifesto [2], which represents the foundation from which this future is built. Instructed by Birra Gubba and Gungalu elder and academic Dr Lilla Watson’s call, we envisage a future of health and humanity that is only made possible via Indigenist terms of reference, as first articulated by Narungga, Kaurna, and Ngarrindjeri scholar Professor Lester Rigney [3]. In making the case for Indigenist Health Humanities we briefly chart the inglorious history of health research in the ‘Australian’ Indigenous context, from its past silence to its present failures, and chronicle the emergence of a new field of research that will advance the knowledge required to attend to the persisting health inequality experienced by Indigenous peoples. In starting with the story of health inequality, we could well be accused of establishing a field on the same troubled foundations of deficit and despair that most operate on, reproducing the racialised knowledges that make illness an inherent part of the Indigenous condition. But it is outrage at the injustice of inequality that forms the foundation of this field. It provides us with an imperative to act, in service of the supposed subjects of study, rather than performing the illusion of objective and impartial knowers of their experience. The fetishised search for ‘best practice’ located within conservative positivist research agendas camouflage, rather than address, injustice.

It is rage that fuels the formation of this field.

*I did listen to my rage, allow it to motivate me to take pen in hand and write in the heat of that moment. At the end of the day, as I considered why it had been so full of racial incidents, of racist harassment, I thought that they served as harsh reminders compelling me to take a stand, speak out, choose whether I will be complicit or resist. All our silences in the face of racist assault are acts of complicity. What does our rage at injustice mean if it can be silenced, erased by individual material comfort? If aware black folks gladly trade in their critical political consciousness for opportunistic personal advancement then there is no place for rage and no hope that we can ever live to see the end of white supremacy. Rage can be consuming. It must be tempered by an engagement with a full range of emotional responses to black struggle for self-determination.* (bell hooks [4])

### Bridging the Health Gap via the Knowledge Gap

The epidemiological narrative has become a taken-for-granted feature of Indigenous health discourse in recent decades, demonstrated most notably in the handing down each year of the failing Closing the Gap report to the Australian parliament [5,6] However, these statistical portraits have not brought “the rewards that a scientific imagination might assume” [7] (p. 90), such as a more organised examination of the production of health disparities, including the practices of health research, policy, and service provision; and the social, political, economic, and historical conditions that create and maintain such inequalities. Epidemiology has long been the conjoint of how Australians have come to “know” Aboriginal and Torres Strait Islander people [8]. This “knowing” has painted Indigenous people into a bleak corner of humanity where they are reduced to a continuum of just four possibilities: at risk of sickness, sick, dying, or dead [9]. A discipline so neutral it cannot, or rather will not, account for state-sanctioned violence, oppression, and dispossession in ways that could describe, explain, predict, and control inequity, but instead reinforces whiteness as superior—the norm by which the diseased and deviant Indigenous population will be measured. Inasmuch as the discipline has contributed to highlighting the health inequities experienced by Indigenous peoples, it has created, produced, and perpetuated accounts of ‘alarming’ negative statistical portraits that garner little to no action, as they simultaneously render Indigenous people less than human, and less capable of living a healthy life [10]. Instead, the gap of Indigenous health inequality continues to be a ‘known problem’ with ‘unknown solutions’, attributed to a mysterious failure in resourcing and implementation, despite renewed commitments or refreshed numerical targets.

The National Health and Medical Research Council (NHMRC) has committed 5% of its total budget to Indigenous health research for well over the past decade [11]. This dedicated funding is necessary but remains insufficient as they have yet to invest in a research agenda that has an explicit commitment to transforming health outcomes over its existing preoccupation with describing Indigenous ill-health [12] (p. 2). This increased investment in Indigenous health research has led to increased visibility of Indigenous ill-health as a problem [13], but this has yet to translate to a reduction in Indigenous health disparities at the rate which is so urgently needed. Neither statistically significant nor culturally significant, Indigeneity comes to represent a category of risk in health research, devoid of rights, but in need of being known.

Nurungga scholar Professor Lester Rigney has described the efforts of Indigenous scholars from Australia and the Pacific who, in contesting the ongoing production of racialised knowledges about Indigenous peoples, have sought “more progressive knowledge seeking methods that privilege the diversity of Indigenous experiences” [14]. He explains:

*…I have used the term Indigenism to describe and define this body of knowledge and its discourse…what I mean by this term is a distinct Indigenous Australian academic body of knowledge that seeks to disrupt the socially constructed identity of the ‘archetypal Aborigine’, as a controlled and oppressed being, that informed the emergence of a distinct yet diverse Indigenist Research epistemological and ontological agenda. There are clearly many historical, social and political factors that have led to the emergence of Indigenism in Australia…However, it is important to note that classic scholarly works of anticolonialism have provided valuable theoretical approaches to the contemporary Indigenous Australian Intellectual Movement in its interrogation of dominant research tendencies that assume central positions of ‘power’ and ‘truth’. Moreover, the debates in Aotearoa (New Zealand), the United States and Canada by First Nations peoples, scholars of color, and non-Indigenous people have also influenced methodological reform developed by Indigenous peoples in the Pacific. Australian Indigenism has also capitalized on new and flexible ways to conduct research.* (Lester Rigney [14] (pp. 37–38))

## 2. Building a Community of Practice on Indigenist Terms

Indigenist Health Humanities seeks to bridge the knowledge gap of Indigenous health by broadening the intellectual investment: inviting humanities and social science perspectives about the social world that Indigenous people occupy to better understand its role in the production of health, illness, and inequality. This is particularly salient given the increasing recognition of the social and cultural determinants of health, both locally and globally [15,16]. The assertion of an ‘Indigenist’ health humanities, as opposed to the emerging fields of medical and health humanities, is an important demarcation that recognises the violence of the humanities upon Indigenous peoples. Indigenist Health Humanities makes explicit the criticality of critical Indigenous studies and, particularly, Rigney’s Indigenist research principles of resistance, political integrity, and privileging of Indigenous voices [14]. Indigenist Health Humanities insists upon a foregrounding of Indigenous intellectual sovereignty to resist and remedy the prevailing racist research paradigms found across both health and humanities. Similarly, Indigenist Health Humanities is not a field whose parameters are defined by the Indigeneity of researchers or research subjects; rather, it is a field that regards Indigenous knowledges as foundational for knowing not just an ancient past, but a possible future. In being Indigenist, rather than Indigenous, neither the knowers or known must be Indigenous; however, the principles of Indigenist research, as expressed by Rigney, provide the parameters by which knowledge is produced.

Indigenist Health Humanities as a field of research harnesses a holistic and reparative methodology in the context of Australian health research. It represents a new Indigenous health research paradigm that can revitalise efforts to improve health beyond an Indigenous Australian context. The parameters for this field emerged from a one-day workshop convened in Inala, Brisbane (4 June 2019) among an interdisciplinary team of Indigenous and non-Indigenous scholars interested in expanding the current parameters of Indigenous health research from within and beyond their disciplines. From this workshop emerged the Inala Manifesto, which was first launched at the Lowitja Institute’s International Indigenous Health Conference in Darwin, Northern Territory [2]. The Inala Manifesto represents six foundational values that provide the framework for forging a new field of research. Thus, Indigenist Health Humanities is research which:Recognises that the prevailing Indigenous health research paradigm is an apparatus of colonial control and thus calls for a resistance against the ideological foundations which insist that ill-health experienced by Indigenous peoples is a product of Indigenous deficit, biologically or culturally.Foregrounds Indigenous intellectual sovereignty, recognising the centrality of an Indigenous criticality to Indigenous health advancement which is defined by Indigenous peoples, making visible the strength, capability, and humanity of Indigenous peoples in all processes and products.Recognises health and wellbeing as a fundamental human right, including recognition of Indigenous peoples’ unique rights as articulated via the United Nations Declaration of the Rights of Indigenous Peoples [17]. As Indigenous rights are highly contested, health is a matter of justice and health research too becomes a question of politics and political struggle, rather than simply the production of an evidence base for action.Attends to the nature and function of race within an Australian context and not just in how Indigenous people experience racism in the health system, but, more broadly, how race operates in the everyday, from birth to death, including the embodied consequences of racism.Demands courage of health researchers to shift the gaze away from Indigenous incapability to consider how institutions, structures, systems, and processes operate to undermine Indigenous health and wellbeing. Consequently, it is also concerned with developing the researcher’s toolkit beyond the academy as a public intellectual and change-maker.Considers disciplinary disloyalty a form of academic excellence, demanding a shared allegiance to Indigenous health advancement and action rather than adherence to disciplinary knowledges and power. In this instance, the sociologist, the anthropologist, the political scientist, the legal scholar, and the philosopher are as integral to the health research team as the clinically trained health researcher and epidemiologist. It is precisely with the collective coming together on the terms articulated here that new conceptual tools and frameworks can be brought to bear to carve out the necessary landscapes for imagining and enabling Indigenous health and wellness.

In its commitment to Indigenous advancement, this field of research is configured around mobilising an Indigenous intellectual collective capable of effecting the necessary social, political, and economic change required for the achievement of better health while simultaneously building the field of Indigenist Health Humanities. It is inclusive of non-Indigenous scholars and their scholarship but reconfigures, rather than reinforces, existing hierarchical relationships that have traditionally favoured non-Indigenous health researchers. Indigenous health advancement is as much a matter of strategy as it is a matter of knowledges, and a stress on broadening research methods, in combination with Rigney’s principles, is designed to level the western epistemological edifice that presently maintains the status quo.

Indigenist Health Humanities brings together health and the humanities through the particularity of Indigenous scholarship and Indigenous advancement. While health humanities has gained some traction within the academy, it remains removed from those for whom health outcomes are the worst. An Indigenist Health Humanities agenda would not emulate but disrupt the abiding, yet less than inclusive, understandings of what the academy and wider public health discourse take health humanities to be. Broadening the focus of health research in this way requires embracing interdisciplinarity: marrying disciplines in a bid to fully encompass the Indigenous human experiences of health. By extension, an ideal health research team in this regard would bring together the disciplines of applied philosophy, the social and political sciences, legal and literary studies, and history alongside those of epidemiology and the clinically trained health researcher—which is exemplified in our investigative team. Through this work, the categories of health research and health researcher will be widened, thus also broadening the possibilities for current Indigenous health research investments beyond the existing individualised, medicalised, deficit-based, capacity-building approaches. In foregrounding Indigenous intellectual capabilities through the deliberate cultivation of an intellectual collective, a more sustainable, relational, and ethical approach to advancing new knowledge that advances health outcomes for Indigenous peoples will emerge.

Edward Said, Palestinian public intellectual and founder of post-colonial studies, described the opportunities that arise from the challenges of transformative scholarly work:

*For the intellectual an exilic displacement means being liberated from the usual career, in which ‘doing well’ and following in time-honoured footsteps are the main milestones. Exile means that you are always going to be marginal, and that what you do as an intellectual has to be made up because you cannot follow a prescribed path. If you can experience that fate not as deprivation and as something to be bewailed, but as a sort of freedom, a process of discovery in which you do things according to your own pattern, as various interests seize your attention, and as the particular goal you set yourself dictates; that is a unique pleasure... The exilic intellectual does not respond to the logic of the conventional but to the audacity of daring, and to representing change, to moving on, not standing still.* (Edward Said) [18] (pp. 62–64)

## 3. The Vision

Taking the core principles articulated via the Inala Manifesto [2], as a collective we have taken up the ‘audacity of daring’ to carve out core streams we believe underlie the formation of Indigenist Health Humanities as a new field of research. This new field provides an intellectual home for the exiled intellectuals whose work is committed to the “special duty” of addressing “the constituted and authorised powers of one’s own society” [18] (p. 98). That being said, such a home is open to expansion to accommodate the increasing members who find warmth and solidarity in the refusal of injustice that binds us intellectually and politically to such a place. Inasmuch as this is an intellectual project, the political nature of it demands a strategising that attends to, rather than acts in spite of, the forces of marginalisation that such scholars are subject to—particularly when such scholars are not likely to be beneficiaries of institutional patrons. This field is formed through the engagement of disparate researchers, research units, and institutions, not working in isolation but working together in a more collective fashion. The foundations here have emerged from such relationships among the authors and are outlined below.

### 3.1. Indigenous Knowledges as Pedagogy

Led by Drummond, this stream explores the contribution of Indigenous knowledges to contemporary health discipline scopes of practice, specifically in the education preparation of health professionals. Centred upon the scholarship of teaching, Ali Drummond’s PhD and forthcoming postdoctoral research will further engage and theorise Indigenous ways of knowing, being, and doing, as embodied and emplaced by Indigenous health academics, as essential to nurturing a health workforce that will primarily enhance their work with and for Indigenous peoples. This scholarship will implicitly explore and promote regulatory mechanisms for the use of Indigenous knowledges as pedagogies and curricula content. This necessitates the expansion of the current community of practice for Indigenous health scholars and higher degree research students to rightfully encompass their communities with the express goal of upholding Indigenous intellectual sovereignty within health discipline education.

### 3.2. Philosophising Health as Life

The dominant approach to philosophy across Australian universities presumes that meaningful thought originates elsewhere, typically from the Anglo-American tradition or continental Europe. The effect is the perpetuation of what Rigney terms ‘intellectual nullius’ [19]. An Indigenist approach to philosophy on the other hand—an Indigenist approach to the pursuit of wisdom, the creation of concepts, and the shaping of direction and procedures employed in human and social sciences—begins with the critical and creative intellectual works and capacity of Indigenous peoples. In conversation with Mukandi, a community of thinkers including community leaders will gather to study critical and creative Indigenous affirmations of health and life; engage in dialogue and debate; create concepts; and work towards writing the beginnings of a canon of Indigenist philosophy of health.

### 3.3. An Indigenist Epidemiology

While the limitations of epidemiological approaches to Indigenous health are well-documented, an Indigenist epidemiology takes on the challenge of Rigney’s Indigenism [3], interrogating the possibility of a transformative epidemiology which centres the human experience of Indigenous peoples—culturally, politically, intellectually, and ethically—throughout all stages of research, including data collection, analysis, and translation. Led by Whop, an Indigenist epidemiology will emerge that is committed to driving change over performances of objective observations. Indigenist epidemiology will stretch the parameters of existing attempts at strength-based approaches within the discipline to foreground Indigenous sovereignty and survival, including more meaningfully accounting for race that does not continue to render Indigenous peoples statistically insignificant or inferior.

### 3.4. Unsettling Colonialism

This stream seeks to apprehend and reveal the violence of colonial systems and institutions through which health is currently conceived and managed, requiring an expanded conceptual language where the frame for ‘health research’ moves beyond ‘evidence-based policy and practice’ to political dynamics and relations, as well as to questions of redress and transformation. Drawing upon the expertise of Brough and Macoun, this stream brings to bear the social and political sciences not simply in understanding how the health system works, but how various macro political, economic, and social structures operate in the production of health inequalities. Critical of the dominant individualised, behaviourist discourse, this scholarship includes and extends theoretical and applied understandings of settler colonialism and the social determinants of health, bringing together critical social researchers from justice, law, health, education, and creative industries to take up a transformative agenda which pushes, rather than privileges, orthodoxies as a catalyst for social change. This work necessarily involves destabilizing settler presence, structures, and futures in order to contribute to the project of ensuring the intellectual and political authority of Indigenous peoples is respected and taken for granted, rather than contested.

### 3.5. Indigenous Critical Race Theory

Critical race theory (CRT) traces its origins to legal studies in the US but it has yet to take seriously the intersection of race and Indigeneity or foreground the intellectual sovereignty of Indigenous peoples in the global south. An Indigenous critical race theory extends on Watego and Singh’s work which seeks an Indigenous public health application for CRT that centres the embodied experiential knowledge of Indigenous peoples. The goal of an Indigenous critical race theory is to imagine Indigeneity not as a variation to being human but as foundational, and, in doing so, to illuminate understandings of what it is to be uniquely and fully human in relation to the natural and ancestral world. This work seeks to forge a Black (When referring to Aboriginal and Torres Strait Islander peoples, we use varying terms such as Indigenous and Blackfullas. Indigenous can include First Nations peoples locally and globally, while the term Blackfulla is one of many terms used by Indigenous peoples within so-called Australia to refer to themselves and captures the unique intersection of race and Indigeneity) humanity explored and defined by Blackfullas, challenging received notions of liberation and the traditional role of race in animating liberation struggles. This includes critically examining the purpose of race scholarship beyond a purely theoretical framework, to include a framing of an ethics of anti-racist practice that foregrounds Indigenous sovereignty. It also offers a deeper understanding and delineation of how Blackfullas have imbued meaning to race. It seeks development of a critical mass of Blackfulla race scholars through an innovative teaching program that brings Blackfullas, Black communities, and victims of racial violence together in company and righteous rage. It seeks the creation of spaces—culturally, intellectually, and politically—for those negatively racialised to speak freely about race and how it makes and breaks them, but also to strategise how to make the perpetrators of racial violence pay.

### 3.6. Health Justice

Led by Newhouse and Kajlich, health justice refers to scholarship that understands issues of justice as not only located in colonial legal systems, but through and because of Indigenous sovereignty. Kajlich’s research, which includes unprecedented access to the National Coronial Information System, has examined the varying ways that race operates in health and legal systems in order to inform better strategies for health justice. This scholarship will extend and enable the current work of the National Justice Project, providing the intellectual rigour for legal responses to the ongoing health injustices Indigenous peoples experience. Through this research, justice is understood not only as a strictly legal response or remedy for racial harm in the health system, but as intimately tied to health and the systems and services that care for Indigenous peoples because they are grounded in place, community, and action. It also involves intellectual and political strategising—in relation to law reform—locally and globally, recognising that colonial legal systems must be challenged and held to account for their overt and insidious failures. This is particularly critical in relation to the ongoing refusals to attend to the harms of race and racism beyond narrow definitions of racial discrimination and the limited, and often problematic, remedies currently available: remedies which were never designed to deliver health justice for Indigenous peoples or, more fundamentally, that could imagine health justice as foundational to the wellbeing of Indigenous peoples.

### 3.7. Transformative Knowledges

Led by Indigenous journalist and social commentator McQuire and health researcher Stajic, this stream recognises the centrality of social transformation to knowledge production and the role of traditional and new media and the creative industries as enablers of achieving such objectives. McQuire’s current work critically examines media reporting of violence against women, and, in response, she has developed her own methodology for humanising women as part of this work. Furthermore, Stajic’s work incorporates digital storytelling to amplify Indigenous theorising about health and healthcare as articulated by Aboriginal and Torres Strait Islander health workers. Here we seek to engage with literary scholars, artists, and activists to explore new methodologies for knowledge production that centre social transformation. Creative works are not simply the means to decorate or communicate the knowledge accomplishments of the health sciences, but are respected here as legitimate forms of knowledge production in their own right, making a critical contribution to understandings of health and humanity.

## 4. Discussion

Indigenist Health Humanities offers the required critical imagination for better understanding of the complexities of the social and the cultural in producing ill-health and/or promoting better health. Through this, we might come to realise the limitations of drawing too heavily upon a medical response to what is effectively a socio-political problem, enabling us to extend our strategies for health advancement beyond individual illness and health behaviours to include attending to the social, economic, cultural, legal, linguistic, and political conditions in which health inequalities are produced and maintained. It enables a shift in focus for researchers from documenting and attempting to address inequalities between Indigenous peoples and a non-Indigenous norm, to creating the conditions through which Indigenous peoples’ understandings and sovereign expressions of health and wellbeing might be realised, which extend beyond the narrow parameters of closing statistical health gaps.

Indigenist Health Humanities offers a way to break out of the biomedical mould that has struggled to encompass the life worlds of Aboriginal and Torres Strait Islander peoples, and offers instead an original way of asking new questions of ‘old problems’, as well as contesting the very construction of these problems. It reconfigures Indigenous peoples from a problem to be solved to that of knowledge bearers of both strategy and solutions for survival. While the problem of Indigenous ill-health provided the impetus for a new imagining, the application of Indigenist Health Humanities is not confined to Indigenous peoples. It is the criticality of Indigenous studies that is being brought to bear to broaden our imaginings of health and humanity. Indigenous peoples are not a subset of a population group, rather, Indigenous sovereignty is the foundation from which a new future can be conceived—one that is most sustainable, most equitable, most caring, and most humane.

What is most innovative about this field is the recognition of Indigenous sovereignty in the formation of inquiry that has a global application in relation to the amorphous field(s) of medical and health humanities, which represent an academic configuration still in formation [20,21]. While some may see these as interchangeable, others view the ‘health humanities’ to be an advance over the ‘medical humanities’, one which resists the tendency towards an overemphasis on the biomedical [22,23,24]. This concern, of course, precedes the development of the health humanities, with the longstanding supposition that the incorporation of the humanities into medical education makes for more rounded, humane medical practitioners [21,25,26]. An Indigenist Health Humanities extends the aim of deepening “our understanding of human health and wellbeing by calling on multiple perspectives—biomedical, philosophical, historical, artistic, literary, anthropological and sociological” [27] (p. 5); to actors beyond the clinician as sole teacher, and clinician as central figure within the health humanities [22,24]. If people are understood in their full, relational sense rather than as atomistic, potential loci of pathology, their health “is reconceptualized as something that is produced through the relations between bodies rather than as something that a body is or is not” [23] (p. 77). The challenge, as well as the most lucrative site of investment across the health humanities via an Indigenist Health Humanities, is the study of those relations. Put another way, a key task that lies before the health humanities is to begin to address the full humanity of members of marginalised groups. While there have been attempts at articulating an ‘Indigenous health humanities’, in its failure to foreground Indigenous sovereignty, Indigenous knowledges and peoples remain marginalised [28].

Indigenist Health Humanities is a field of research that disavows Black lack and instead marshals various methods commonly used in the humanities (e.g., film making, literary criticism, creative writing, ethnography, etc.) to investigate the ways in which power is configured to the detriment of Indigenous peoples’ health and wellbeing. It is a field of research that recognises that Indigenous ways of working based on relationality, as opposed to the neo-liberal valorisation of individuality and leadership. We offer a new way of thinking about health and Indigeneity and, through the Inala Manifesto, offer an alternative vision of scholarship and knowledge production as a collective endeavour guided by Rigney’s Indigenist principles. Through this investment, we will also advance understandings of research impact beyond the narrowly defined parameters which it is currently measured against.

## 5. Conclusions

As the oldest continuous culture on the planet, Aboriginal and Torres Strait Islander peoples have much to teach about survivance in a rapidly changing world. To think globally through Indigeneity is to return to a future that foregrounds Indigenous sovereignty, sustainability, and relationality. These principles can guide the transdisciplinary turn which is still firmly wedded to conceptions of modernity and change—the very ideas that have generated some of the global challenges we face, not just those uniquely experienced by Indigenous peoples. The transdisciplinarity required to effect change requires more than a bringing together of different methodologies—it demands attention to different ways of knowing and being in a relational, rather than hierarchical, manner, recognising the limitations of different knowledge systems as well as their strengths, so that the most appropriate conceptual tools are brought to bear in addressing the grand challenges we face both now and into the future.

## Data Availability

Not applicable.

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
