# Peer review of "Black to the Future: Making the Case for Indigenist Health Humanities"

_ijerph, 2021, doi:10.3390/ijerph18168704_

Round 1

Reviewer 1 Report

Congratulations! While I have added several comments for consideration, overall, I think it is an excellent piece.

Author Response

Please see attached table in document. 

Reviewer 2 Report

Title: Black to the Future: Making the Case for an Indigenist Health Humanities

Thank you, IJERPH Editors, for the opportunity to review this manuscript.

Thank you, also, to the Authors – this is a Mighty piece of work, which has the potential to add to the literature in very interesting and important ways.  I am particularly impressed by the “righteous rage” (318) that motivates this piece – that is, “It is rage that fuels the formation of this field” (71); “outrage at the injustice of inequality” (65-66).  Fantastic work, Authors – I look forward to referencing this article within my future work.

My main critiques of this draft manuscript relate to: adequacy of Referencing; consistency of terminology (e.g., "research agenda" OR "field of research" OR "research paradigm" OR "research program" OR "methodology"); consistency of capitalising Indigenist Health Humanities; suggestions for some additional words (sentences, diagrams, etc.) to make the argument clearer and more accessible for a diverse (international) readership; and a suggestion to split the current Conclusion section into two (e.g., Discussion, Conclusion).  Authors will find my specific suggestions highlighted and noted within the attached PDF.

Again, congratulations, Authors: I am very impressed by your Vision, as well as the work being undertaken by your team.  Keep up the excellent work.

Author Response

Please see attached table in document

Reviewer 3 Report

while I have ticked the above categories such as research design - the argument for writing this call for an ethical response was clear.  Thanks.

I teach in critical space, including the engagement of decolonising western knowledge assumptions of deficit through sameness and difference - and i was moved by this collective narrative - it is transformational.

Author Response

Please see attached table in document. 
